

**Effect assessment of $NO_x$ and $SO_2$ control policies on acid species in precipitation from 2005 to**
**2016 in China based on satellite monitoring**
Xiuying Zhang [1, *], Dongmei Chen [2, 3], Lei Liu [1, 4], Limin Zhao [1], Wuting Zhang [1]
[1] International Institute for Earth System Science, Nanjing University, Nanjing 210023, China
[2] Department of Geography and Planning, Queen's University, Kingston, ONK7L 3N6, Canada
[3] School of Geography and Remote Sensing, Nanjing University of Information Science & Technology,
Nanjing, China
[4] Jiangsu Center for Collaborative Innovation in Geographical Information Resource Development and
Application, Nanjing 210023, China
* Corresponding authors: Xiuying Zhang (lzhxy77@163.com)
**Abstract**
The effects of $NO_x$ and $SO_2$ policies on acid species in precipitation was assessed in China from 2005
to 2016, based on the OMI measured $SO_2$ and $NO_2$ columns. The results showed that the $SO_2$ and $NO_2$
columns in the atmospheric boundary layer (ABL) could be used to indicate the variations of S / N in
precipitations (R = 0.90, incept = 0.97, P < 0.05). The spatial distribution of S / N was lower in eastern
China than the west, which had a negative logarithmic relationship with population densities (R = 0.78,
P < 0.05). The OMI-derived S / N decreased significantly from 2005 to 2016 (17.21 and 10.70 in 2005
and 2016, respectively), mainly due to the controlling S and N policies enacted at different times. The
ABL $SO_2$ columns showed a decreasing trend from 2005 to 2016, while $NO_2$ presented an increasing
tendency from 2005 to 2011 then decreasing until 2016. The temporal variations of $SO_2$ and $NO_2$ were
not only determined by their emissions but also affected by precipitation amount, which induced the
highest $SO_2$ and $NO_2$ concentrations in 2011 during the study time. With the combined acidification



effects of S and N, the acidity had increased from 2005 to 2011, then decreased until 2016. The acidity
in 2016 has declined 11.0% and 25.4%, respectively, compared with those in 2005 and 2011, indicating
the policies on joint controlling $SO_2$ and $NO_2$ have gained effects.
**Keywords**: Acid species, precipitation, policies, China, remote sensing, OMI
**1. Introduction**
The economic growth in recent 30 years in China has been accompanied by increased energy demand
and massive emissions of pollutants (Kuribayashi et al., 2012). Among them, sulfur dioxide ($SO_2$) and
nitrogen dioxide ($NO_2$) are important precursors of acid rain. After released into the atmosphere, $SO_2$
and $NO_2$ can be transformed into nitrate ($NO_3^-$) and sulfate ($SO_4^{2-}$) through complex physical and
chemical processes (Yu et al., 2016), and then diluted with precipitation. Increased acid deposition and
the followed decreased pH of precipitation, directly and indirectly, influence the eco-environments
(Charlson and Rodhe, 1982;Li et al., 2017;Larssen and Carmichael, 2000). Currently, China is
becoming one of the highest acid deposition areas on a global scale (Vet et al., 2014), which has been
drawing serious concerns from the public and policy makers in China due to the adverse impacts of
acid rain on the ecological environment.
To control acid rain pollution, the Chinese government has implemented a series of policies from the
$9^{th}$ to $12^{th}$ Five-Year Plans (1996-2015). "Decision of the State Council on Several Issues Concerning
Environmental Protection" was issued in 1996, in which "15 major categories of small pollutant
enterprises" related to pollutant emissions should be closed ([1996]31). In 1998, the Chinese
government adopted national legislation, known as the "two control zones (TCZs) plan", to limit
ambient $SO_2$ pollution and halt the increase of acid rain. During the $11^{th}$ Five-year Plan, a clear aim was
announced that national wide $SO_2$ emission in 2010 should be reduced by 10% of that in 2005, and the



increasing tendency of $NO_x$ should be controlled. Under such strict policies, small power generating
units and inefficient industrial facilities were closed, and 82.6% of the thermal power plants were
equipped with flue gas desulfurization (FGD) in 2010 (China Environmental Bulletin 2010). However,
the ground measurements of precipitation pH indicated that the control policies had not been successful
in reducing acid rains (China Environmental Bulletin 2010). Thus several studies have argued whether
it is enough to control acid rain pollutions through only controlling $SO_2$ emissions (Zhao et al.,
2009;Fang et al., 2013).
To further improve the air quality and control acid rains, the Chinese government issued "Twelfth
Five-Year Plan for National Environment Protection" in 2011, and set a goal to reduce the emissions of
$NO_x$ by 10% and $SO_2$ by 8% in 2015. Additionally, the Ministry of Environmental Protection, National
Development and Reform Commission, and Ministry of Finance jointly issued "12[th] Five-Year Plan on
air pollution prevention and control in Key regions" in 2012, which aims to reduce the emissions of
$SO_2$ and $NO_x$ by 12% and 13%, and the concentrations of $SO_2$ and $NO_2$ by 10% and 7% in the key
regions in 2015. Selective catalytic reduction (SCR) equipment was recommended to install in this
period and the SCR equipment number grew from a percentage of about 18% in 2011 to 86% in 2015
(Liu et al., 2016a). Moreover, several new national emission standards for cars have been implemented
to reduce the $NO_x$ emissions generated by traffics (Wu et al., 2017).
Several studies have discussed the effect of environmental protection policies on the emissions,
concentrations, and depositions of S and N in China, based on the ground measurements or satellite
monitoring information (Kuribayashi et al., 2012;Krotkov et al., 2016;Zhao et al., 2013;Ronald et al.,
2017;de Foy et al., 2016). It was found that $SO_2$ emissions in China increased continuously before 2006,
and declined after 2006 due to wide application of flue-gas desulfurization (FGD) in power units since



2005 (Fang et al., 2013;Duan et al., 2016). $NO_x$ emission was also found to have a rapid increase from
1990 to 2011 and decreased from 2012 because of the broad applications of SCR in coal-fired power
plants (Wang et al., 2014). The remotely sensed $SO_2$ and $NO_2$ columns also showed that FGD had a
significant effect when a much stricter control of the actual use of the installations began in the years of
2008-2009 (Ronald et al., 2017). The remotely sensed $NO_2$ columns reached the peak in 2011 and then
decreased from 2012 in China (Chen et al., 2017). But the emission peak years for $NO_2$ varied from
province to province (Ronald et al., 2017). Although the peak years for $SO_2$ and $NO_2$ varied in different
studies, these studies confirmed that atmospheric S and N increased first due to the economic
development, and then decreased because of the enacted policies.
The different changing steps of S and N in the atmosphere, including the different increasing /
decreasing rates and peak years, directly influence the S and N depositions through precipitation
(Huang et al., 2009;Liu et al., 2016d;Liu et al., 2016c). According to a review on chemical
compositions precipitation in recent years, the pollution in precipitation is still dominated by sulfur
type with a trend to sulfuric-nitrous mixed type in China (Wang and Xu, 2009). The declined ratio of
$SO_4^{2-}$ to $NO_3^-$ concentration has been observed in northern China (Wang et al., 2012;Pu et al., 2017),
southeast China (Huang et al., 2008;Fang et al., 2013), and southwest China (Liu et al., 2016b),
indicating the proportion of $NO_3^-$ concentration on precipitation acidity has been increased in
recent years. Combining the acidification effects of S and N, the benefits of $SO_2$ reductions during
2005−2010 might be largely offset by increases of N emissions (Zhao et al., 2009). However, the
change of the contributions of $SO_4^{2-}$ and $NO_3^-$ on precipitation acidity has not been evaluated,
particularly on a national scale.
Compared with the ground measurements, the remote sensing technique provides a new means to





monitor the concentrations of $SO_2$ and $NO_2$ in the atmosphere, with the advantages of extensive spatial
details and continuous temporal coverage (Zyrichidou et al., 2013). At present, the Global Ozone
Monitoring Experiment (GOME2A, 2007-; GOME2B, 2013-), Scanning Imaging Absorption
Spectrometer for Atmospheric Cartography (SCIAMACHY, 2002-2012), and the Ozone Monitoring
Instrument (OMI, 2004-) on the Aura satellite provide both of the $SO_2$ and $NO_2$ column products.
Among them OMI provides the smallest instantaneous ground pixel resolution ($13 \times 24$ $km^2$ at nadir)
and its products have been widely used for studying the spatial and temporal variations of $SO_2$ and $NO_2$
concentrations in the atmosphere and evaluating the effects of policies (Ronald et al., 2017;Chen et al.,
2017;Krotkov et al., 2016;Song and Yang, 2014;de Foy et al., 2016;Lamsal et al., 2015). However, few
studies have used the remotely sensed $SO_2$ and $NO_2$ columns data to indicate the variations of acid
components in precipitation (Liu et al., 2017;Zhang et al., 2012).
Remotely sensed $SO_2$ and $NO_2$ columns have been used to study the effect of precipitation variation on
severe acid rains in southern China (Xie et al., 2009), and evaluate the influence of $SO_2$ and $NO_2$
columns on precipitation pH in Anhui and Liaoning Provinces (Shi et al., 2010;Zhang et al., 2012).
More particularly, Liu et al. estimated monthly $NO_3^{-1}$ deposition through precipitation using the OMI
$NO_2$ columns and precipitation amount (Liu et al., 2017). In their work, the atmospheric boundary layer
(ABL) $NO_2$ columns below the precipitation height instead of the tropospheric $NO_2$ columns were used
to construct the statistical model to estimate $NO_3^{-1}$ depositions, since the scavenging effect on N
compounds is from the top precipitation height rather than the top troposphere height (Racette et al.,
1996). Therefore, the ABL $SO_2$ and $NO_2$ columns should be better than tropospheric columns when
indicating the acid components deposited.
Based on the ABL $SO_2$ and $NO_2$ columns, this study aims to evaluate the trends of the contributions of





111 the sulfate and nitrate ions on the acidity of precipitation from 2005 to 2016, under the policies enacted

112 and economic development in China. First, the remotely sensed indicator of S / N to the ratio of sulfate

113 and nitrate ions in rainwater is constructed based on ABL $SO_2$ and $NO_2$ columns; second, the spatial

114 variations and the trends of the S / N in rainwater are detected; finally, the spatial and temporal

115 variations of the potential acidification by sulfate and nitrate ions are evaluated.

116 **2. Materials and methods**

117 **2.1. Materials used in this study**

118 **2.1.1. Tropospheric $NO_2$ and ABL $SO_2$ columns from OMI**

119 The OMI satellite instrument is a nadir-looking UV-visible spectrometer on the Aura satellite (Levelt et

120 al., 2006). Aura was launched on 15 July 2004 and flies in a sun-synchronous polar orbit with a local

121 equatorial overpass time of 13:40 on the ascending node. The $NO_2$ columns are provided in the publicly

122 released level 2.0 (DOMINO 2.0) (http://www.temis.nl/), which has greatly improved the accuracy of

123 the tropical $NO_2$ columns of the version 1.0 (Boersma et al., 2011). A new data set of ABL $SO_2$

124 columns from OMI is available, which agree on average within 12% with ground observations,

125 strongly improved on earlier $SO_2$ data sets from satellites (Theys et al., 2015).

126 In this study, the monthly tropospheric $NO_2$ columns from Jan 2005 to Dec 2016 over China are used.

127 The missing data is interpolated using IDW (Inverse Distance Weight) method, and then the unit of DU

128 (Dobson unit) for $SO_2$ column is transformed to molec. $cm^{-2}$ by multiplying $2.6875 \times 10^{16}$ molec. $cm^{-2}$

129 (1DU = $2.6875 \times 10^{16}$ molec. $cm^{-2}$) (Lee et al., 2011). Finally, the monthly mean $SO_2$ columns are

130 averaged from the daily data.

131 **2.1.2. $NO_2$ profiles simulated from MOZART-4**

132 The 56 levels of $NO_2$ concentrations in the atmosphere along altitudes from 2005 to 2016 have been




simulated from MOZART-4, a global chemical transport model. This model is driven by NCEP/NCAR
reanalysis meteorology and uses emissions based on POET (Precursors of Ozone and their Effects in
the Troposphere), REAS (Regional Emission inventory for Asia) and GFED2 (Global Fire Emissions
Database, version 2). Evaluation with several sets of observations shows that MOZART-4 can
reproduce tropospheric chemical composition with an acceptable accuracy (Emmons et al., 2010). The
output data used in the current work are temporally varying six hours every day, which are upon
request by Louisa Emmons at National Center for Atmospheric Research (NCAR)
(http://www.acom.ucar.edu/wrf-chem/mozart.shtml).
**2.1.3. Concentrations of $SO_4^{2-}$ and $NO_3^-$ in precipitation during 2005 - 2016 collected from the**
**published papers**
To test whether OMI-derived S/N could be used to indicate $SO_4^{2-}$ / $NO_3^-$ in precipitation, the $SO_4^{2-}$ and
$NO_3^-$ concentrations in precipitation during 2005-2016 in China from the published studies were
collected. The detailed information on searching the relevant papers has been detailed described in
those studies (Liu et al., 2016c;Liu et al., 2016d). Since the mentioned two studies collected the data
from 2000-2013, we removed the data from 2000 to 2004 and added the $SO_4^{2-}$ and $NO_3^-$ data published
during 2014-2016 in this study. In total, 168 records on annual $SO_4^{2-}$ and $NO_3^-$ concentrations in
precipitation have been selected.
**2.1.4 Spatial distribution of population density**
The spatial distribution of population density in 2010 in China is used to study the relationship between
the S / N and population density. The spatial resolution is 1 km × 1 km. This data set was produced by
integrating the physical geography factors and the statistical data on population in 2010, which is freely
downloaded from http://www.geodoi.ac.cn/doi.aspx?doi= 10.3974/geodb.2014.01.06.v1.



**2.2. Methods**
**2.2.1. Calculation of ABL NO$_2$ columns**
The MOZART-4 output of NO$_2$ concentrations includes 56 vertical levels from the ground to the top of
the troposphere. To simulate the profile of NO$_2$, a Gauss function is used:
$\mathrm{f}(C_h^M) = \sum_{r=2}^{n} a^r exp\left(\frac{-(h-b_r)^2}{c_r^2}\right)$     (1)
where $C_h^M$ is the NO$_2$ concentrations at the atmospheric height $h$; $a$ refers to the amplitude, $b$ is the
centroid (location), $c$ refers to the peak width, n is the number of peaks to fit. In this study, the models
with n from 2 to 6 are simulated, among which the model with the lowest RMSE (Root Mean Square
Error) and the highest R$^2$ is selected.
The tropospheric and ABL NO$_2$ columns $\left(\Omega_{trop}^M, \Omega_{ABL}^M\right)$ simulated from MOZART-4 are simulated by
an integration method:
$\Omega_{trop}^M = \int_0^{trop} f(C_h^M)$   (2)
$\Omega_{ABL}^M = \int_0^{ABL} f(C_h^M)$   (3)
Then the OMI-derived ABL NO$_2$ column $(\Omega_{ABL}^O)$ is calculated as:
$\Omega_{ABL}^O = \Omega_{trop}^O \times \frac{\Omega_{ABL}^M}{\Omega_{trop}^M}$   (4)
Where $\Omega_{trop}^O$ is the NO$_2$ columns retrieved from OMI.
**2.2.2. Calculation of the OMI-derived S / N to indicate the SO$_4^{2-}$ / NO$_3^-$ in precipitation**
In the study of Liu et al. (2017b), the NO$_3^-$ deposition in precipitation ($D_{NO_3^-}$) could be estimated by the
following equation:
$D_{NO_3^-} = \alpha + \beta(\Omega_{ABL,N}^O \times P - \varepsilon)$     (5)
Where  $\alpha$ and  $\beta$  are the intercept and the slope of the constructed model,   $\varepsilon$  is the site bias, $P$ is the
precipitation amount, and $\Omega_{ABL,N}^O$ is an indicator of N compounds in the atmosphere. Here, $\Omega_{ABL,N}^O$





refers to the ABL NO$_2$ columns.
Similarly, the SO$_4^{2-}$ deposition in precipitation could be estimated by the similar equation format as Eq.
(5)[1]. Here, we directly use the coefficients of β for N and S in the two studies to indicate the dilution
rates on N and S by precipitation, in which  β for N and S was 9.33 and 7.10, respectively (Liu et al.,

181    2017).

Thus the OMI-derived S / N is calculated by :
$S / N = \frac{\beta_S \times \Omega_{ABL,S}^O}{\beta_N \times \Omega_{ABL,N}^O}$     **(6)**
The correlation coefficient between the OMI-derived S / N and the collected data of *SO42- / NO3-* in
precipitation is calculated to determine the degree to which the two data sets are associated. Other
parameters of relative error (*RE*) and absolute error (*AE*) are used to assess the accuracy of the
estimated NO$_2$ by the following function:
**2.2.3. Potential acidity induced by H$_2$SO$_4$ and HNO$_3$**
Generally, the precipitation acidity is due to H$_2$SO$_4$ and HNO$_3$, whereas HCl, HF, and other organic
acids are considered as negligible acidity contributors compared to H$_2$SO$_4$ and HNO$_3$ (He et al.,
2010;Khwaja and Husain, 1990). If all of the non-seasalt sulfate and NO$_3^-$ presented in free acid forms,
the potential acidity could be estimated using the sum of nss-SO$_4^{2-}$ and NO$_3^-$ in precipitation (Rodhe et
al., 2002). Since the method and the simulated result on D$_{NO_3^-}$ have been well evaluated by Liu (Liu et
al., 2017), here we directly use the format of the equation to estimate the concentration in H$^+$.
Since H$_2$SO$_4$ has two H$^+$ while HNO$_3$ has one, the latent acidification effects could be calculated as
follows:
$PA = \beta_N \times \Omega_{ABL,N}^O + 2\beta_S \times \Omega_{ABL,S}^O$     (7)

[1] This study has not been published yet.



Here PA is not an actual value of the concentration of $H^+$ induced by $H_2SO_4$ and $HNO_3$, since equation
(7) does not calculate the S and N deposition through $H_2SO_4$ and $HNO_3$. The PA is used here to indicate
the variation of $H^+$ induced by $H_2SO_4$ and $HNO_3$ for a long-term study.
**3. Results and discussions**
**3.1. Validation of the OMI-derived S / N on $SO_4^{2-}$ / $NO_3^-$ in precipitation**
The scatter plots of the OMI-derived S / N and $SO_4^{2-}$ / $NO_3^-$ is illustrated in Fig. 1(a). The OMI-derived
S / N has achieved a reasonably high predictive power on the ratio of $SO_4^{2-}$ to $NO_3^-$ in precipitation,
with a slope of 0.97 and R of 0.90. From Fig. 1(a), four points appatently deviated from the main trend
were located in Xizang, Yunnan, Guizhou, and Hainan provinces (Red circle in Fig 1(b)). If these
points were removed, the R would be increased from 0.90 to 0.94. This confirmed that the OMI derived
S / N could be used to indicate the variations of $SO_4^{2-}$ / $NO_3^-$ in precipitation across China.
The spatial distribution of the relative errors between the collected $SO_4^{2-}$ / $NO_3^-$ and OMI-derived S / N
components is shown in Fig.1 (b). The relative errors ranged from -56.2% to 210.4%, indicating the
ability of OMI-derived indicator on $SO_4^{2-}$ / $NO_3^-$ varied greatly across China. The average of the *RE*
and *AE* was -11.8 % and 22.0 % for the 168 data records, which denoted that the OMI-derived S / N
had underestimated the ratio of $SO_4^{2-}$ / $NO_3^-$- in precipitation. About 134 data records had the *RE* within
-30% to 30%, indicating 80% of the OMI-derived S / N at the collected sample locations could be used
to indicate the $SO_4^{2-}$ / $NO_3^-$ in precipitation. Eighteen and three data records had the *RE* between -45% -
-30% and 30% -45%, respectively. While six data records had very high *RE* values, particularly for the
sites in Southwestern China, which might be caused by the errors of the $NO_2$ and $SO_2$ columns derived
from OMI in these areas.





### 3.2. Spatial distribution of OMI-derived S / N in China

The spatial distribution of OMI-derived S / N in 2016 is illustrated in Fig. 2. The ratio ranged from 0.49

to 71.73, with an average of 10.70, which was much higher than the average of $SO_4^{2-}$ / $NO_3^-$ in

precipitation (2.59) from 474 stations by ground measurements (China Environmental Bulletin 2016).

The big gap was mainly due to that 10.70 was the average of S / N for the whole China, while 2.59 was

calculated from the 474 measuring sites, most of which were located in urban areas. The average of

OMI-derived S / N at the 474 points was 2.78, which was very close to that $SO_4^{2-}$ / $NO_3^-$ in

precipitation in 2016.

According to the classification standard on acid rain types (Cheng and Huang, 1998), 17.3% of the total

areas of China in 2016 had the acid rains of sulfuric-nitrous mixed types (0.50 < S / N < 3). The rest

82.7% had sulfuric acid rains (S / N > 3), among which 49.1% were contributed by the areas with S / N

values higher than 10. Since only several pixels had the S / N values lower than 0.50 (the minimum of

0.49 very close to 0.50), which have been neglected in this study. Thus, the precipitation acidity in

China is still mainly from the contribution of sulfuric species at present.

The large range of S / N indicated that a high variation of sulfate to nitrate components in precipitation

existed across China. The S / N values lower than 3 covered large areas from the northeast China to the

south China, including the whole province of Shandong, Jiangsu, Anhui, and Henan, and partly

provinces of Liaoning, Hebei, Shanxi, Shaanxi, Hubei, Hunan and Zhejiang. Also, in some local areas

around Urumqi, Lanzhou, Yinchuan in northeast China, Chengdu and Chongqing in Central China,

Nanning, Guangzhou and Taiwan in South China, Changchun and Harbin in Northeast China, the S / N

also presented low. Particularly, some areas around Beijing-Tianjin-Hebei, Shandong,

Shanghai-Jiangsu-Zhejiang, Guangdong, Hubei, Shaanxi, and Chongqing-Sichuan, had the S / N less



than 1, indicating the dominant acid component has changed from sulfate to nitrate. While in the
Qinghai-Xizang areas, the northern part of Inner Mongolia, and the south of Yunnan Province, the ratio
showed relative high.
The ratio of S / N was mainly determined by the different sources of $SO_2$ and $NO_x$. In fact, $SO_2$ and
$NO_x$ are released by more or less the same anthropogenic sources, i. e. the burning of coal or oil,
volcanic activity, burning biomass. The main difference to $SO_2$ is that traffic is a much more important
source for $NO_x$ (Ronald et al., 2017). The highly developed traffics in the densely populated areas, and
the more widespread application of the FGD than SCR in power units might be the main reasons for the
low values of S / N in the mentioned regions (Duan et al., 2016;Wang et al., 2014;Fang et al., 2013).
The spatial pattern of S / N is highly negatively correlated to that of population densities (Fig. 2b). To
illustrate the relationship between the population density and the ratio of S to N, the zonal means of S /
N were calculated by the different grades of population density (< 100, 100-200, 200-300, 300-400,
400-500, 500-1000, 1000-5000, 5000-10000 people/km$^2$). It was found that the population density had
a significantly negative logarithmic relationship on S / N (Fig. 3). With the population density increases,
the S / N decreased. The population might not directly influence on the S / N, while it indirectly effects
on the emission source of S and N through human activities.
**3.3. Long-term trends of the S / N from 2005 to 2016**
The averages of the OMI-derived S / N, $SO_2$, $NO_2$ columns are shown in Fig. 4. In China, $SO_2$
concentrations in the atmosphere increased first from 2005 to 2007 and then declined with a little
fluctuation from 2008 to 2016. This trend of $SO_2$ columns is greatly influenced by the variations of $SO_2$
emissions, since $SO_2$ emissions also showed an increasing trend from 2005 to 2006, and then
decreasing until 2016. The consistency of the trends of $SO_2$ emissions and concentrations was



interrupted in 2011 when the SO₂ concentration reached the highest while the SO₂ emission did not.
This inconsistency is mainly because the atmospheric conditions also influence SO₂ concentrations in
the atmosphere through changing the chemical reactions with other components and the vertical and
horizontal transportations (Uno et al., 2007;Emmons et al., 2010;Schaub et al., 2007). Among the
climatic factors, precipitation amount might be a dominant one since it can scavenge the relevant
S-related components in the atmosphere. This study also confirmed that the precipitation amount has an
obvious negative correlation coefficient (-0.69, P < 0.05) with SO₂ concentrations (Fig. 5a). In 2011,
the average precipitation amount was only 556.8 mm in China, which was the lowest in the recent 50
years (China Climate Bulletin 2011). The lowest dilution effect on ions of precipitation in 2011 might
contributed a strong influence on the highest concentrations of SO₂ in the atmosphere. If the year of
2011 was excluded, the correlation coefficient between SO₂ emissions and concentrations in the
atmosphere would have increased to 0.95 from 0.73 (P < 0.05) (Fig. 5b).
The ABL NO₂ columns showed a steadily increasing from 2005 to 2011 and then decreasing until to
2016 (Fig. 3c). Also, NOₓ emissions showed declining from 2011 (China Climate Bulletin 2011, 2012,
2013, 2014, 2015, and 2016), which was highly consistent with the trend of NO₂ concentrations (Fig.
5d). However, although precipitation had the similar effect on NO₂ in the atmosphere with that on SO₂,
precipitation amount did not show a statistically significant negative effect on NO₂ concentrations (Fig.
5c). This might mean that the variations of NO₂ concentrations were mainly determined by the
emissions in China. The maximum of ABL NO₂ columns also occurred in 2011, which might be caused
by both the highest NOₓ emissions and the lowest precipitation amounts.
The possible reasons for the trends of SO₂ and NO₂ concentrations and emissions have been widely
described previously (Ronald et al., 2017;Krotkov et al., 2016), including the economic development



and the implements of a series of policies on atmospheric environmental protection. The trends on $SO_2$
and $NO_2$ directly influenced the variations of acid species in precipitation. The OMI-derived S / N
showed a significant decline (17.21 and 10.70 in 2005 and 2016, respectively), indicating the
contribution of the sulfate on the precipitation pH has decreased while nitrate contribution is increasing.
From 2005 to 2010, the decreasing trend of S / N was due to the slightly decreased $SO_2$ and rapidly
increased $NO_2$. In 2010, $SO_2$ decreased by only 1.0% compared with that in 2005, but decreased by
10.5% compared with the highest $SO_2$ concentration in 2007. During the same period, $NO_2$ increased
by 36.9% in 2010. From 2011 to 2016, both of $SO_2$ and $NO_2$ declined, but $SO_2$ had a higher decreasing
rate than $NO_2$ (43.77 molec. $cm^{-2}$ $yr^{-1}$ for $SO_2$ and 11.39 molec. $cm^{-2}$ $yr^{-1}$ for $NO_2$). The different
decreasing rates induced the further decline of S / N. In 2016, S / N decreased about 37.9% compared
with that in 2005.
The decreasing trend of $SO_4^{2-}$ / $NO_3^-$ in precipitation was also observed from the ground measurements
of the national monitoring on acid rains (China Climate Bulletin 2011, 2012, 2013, 2014, 2015, and
2016), which showed that the ratio has decreased from 3.80 in 2011 to 2.59 in 2016 (3.49, 3.50, 3.18,
2.90 in 2012, 2013, 2014, and 2015, respectively. In some areas of China, such as Jinyunshan, Beijing,
Guangzhou, ground measurements also showed that the $SO_4^{2-}$ / $NO_3^-$ in precipitation decreased in
recent years (Pu et al., 2017;Liu et al., 2016b;Fang et al., 2013).
Considering the classification standard of acid types on precipitation (Cheng and Huang, 1998), the
precipitation acidity is still dominated by the sulfuric species. The percentages of the areas with
sulfuric-nitrous mixed precipitations showed an increasing trend from 9.0% in 2005 to 17.3% in 2016,
while the percentages of the areas with sulfuric rains accordingly has decreased. Particularly, the
percentage of the areas with S / N > 10 showed a much obvious decreasing trend. The average





decreasing rate for the areas with S / N greater than 10 was 1.76 % per year, while the increasing rate
for the areas with S / N less 3 was 0.92% yr$^{-1}$ (Fig. 6). This meant that not only the areas with the
sulfuric-nitrous mixed precipitations sprawled, but also the areas with high S / N values shrank with a
higher decreasing rate. The whole situation confirmed that the contribution of sulfuric species is getting
lower for precipitation acidity in China.
Figure 7 shows the trend of S / N for those grid cells that have a statistically significant trend. A large
negative trend is visible in China, while only several pixels had a positive trend in Taiwan. This is
mainly due to that the environmental regulations on reducing $SO_2$ emissions were implemented earlier
than those on $NO_x$ in China (Ronald et al., 2017). The steadily decreasing trend for $SO_2$ and the first
increasing then decreasing trend for $NO_2$ resulted in the decrease of S / N in most areas, particularly in
east China. The decreasing rate of S / N was within 0-0.25 per year. We should notice that in western
China and the north part of Inner Mongolia, the decreasing rates of S / N were relatively higher than
those in eastern China. This might not be due to the policies on controlling $SO_2$ and $NO_x$ emissions, but
the rapid increased $NO_2$ emissions caused by significant socio-economic changes following the
National Western Development Strategies (The "Go West" movement) (Cui et al., 2016). $NO_x$ might be
getting higher than $SO_2$ due to their different sources, which decreased the S / N rapidly.
**3.4. Long-term trends of the potential acidity in precipitation**
It has confirmed that the $SO_2$ emissions and concentrations in the atmosphere have decreased since
2007. However, the situation of acid rains has not been obviously alleviated since then. According to
the statistic, the ratio of cities with occurring acid rains kept relatively stable around 50% from 2005 to
2014, but rapidly decreased to about 20% in 2015 and 2016 (Fig. S1). The potential acidity (PA) curve
increased first and then decreased (Fig. 8), close to the trend of the ratio of cities with acid rains (R =





0.86, P < 0.05). Even with the year of 2011 not included, this trend did not change but with different
simulated peak years (2008 for 2011 included or 2009 for 2011 excluded). Compared with the highest
PA in 2011, 25.5% of acidic ions have been reduced in 2016. If compared with that in 2005, about 11.7%
of acidic ions have been reduced in 2016, indicating that some successes on the recovery of acid rain
had been achieved.
The spatial distribution of potential acidity in 2005, 2010, and 2016 are described in Fig. 9. From the
map in 2005, the hotspots occurred in north China extending from the northeast China to the Yangtze
delta areas, the highly populated Sichuan Basin, the megacity clusters around Shanghai and Guangzhou.
While in 2010, these regions with high potential acidity still existed. Particularly the areas with
relatively high potential acidity ranged from 10,001 to 15,000 have obviously expanded around Urumqi
in Northwest China. The regions in Xizang and the west Sichuan and Yunnan Province with the
potential acidity lower than 5,000 showed an obvious decrease, indicating the PA had increased.
Combing the decreased ratio of S / N in western China, the acidic ability should be contributed more by
the $NO_3^-$. This is confirmed in the study of $NO_x$ trends in western China (Cui et al., 2016). The contrary
phenomenon was found in Pear River Delta. The potential acidity obviously decreased around the areas
of Guangzhou, which was confirmed by the regional monitoring network data in PRD due to the
deceased $SO_2$ emissions (Wang et al., 2013;Fang et al., 2013).
Although spatial heterogeneities existed between potential values induced by $H_2SO_4$ and $HNO_3$ in 2005
and 2010, the PA in 2010 increased by 2.4% compared with that in 2005. Combing the change of S /N
from 2005 to 2010, the very close PA values in the two years confirmed that the policy only controlling
$SO_2$ emissions had not an obvious effect on alleviating acid rain pollution. A similar conclusion was
obtained through MODELS-3/Community Multiscale Air Quality system (V4.4), in which they




concluded that the benefits of $SO_2$ reductions during 2005-2010 might be negated by increased N
emissions (Zhao et al., 2009).
On the map of potential acidity in 2016, the spatial pattern has changed greatly from those in 2005 and
2010. The hotspots with PA higher than 40,000 are very few, and were mainly located in the northern
China. The big hotspot areas in 2010 occured in Sichuan, Guizhou, and Guangdong Provinces were
gone in 2016, and some small hotspot areas in Ningxia, Hubei, Guangxi, Guangdong were also lost,
meaning that the PA has significantly decreased in these areas. The heavily acid regions extending from
the northeast to the Yangtze areas in 2010 have obviously decreased in 2016. While in the western
China, the PA increased around the Urumqi and in Yunnan, Sichuan and Xizang. If the PA increased
further in the coming years, the acid pollution might be higher in these regions although PA is still low
at present. The increased PA in Western China should be paid attention by the government or the policy
makers.
It should be noted here that the spatial pattern of PA is not consistent with that of the precipitation pH
(Fig S2). The reason for the acidity of precipitation are complex, and many of the primary influencing
factors such as sulfide emissions caused by fossil fuel consumption, atmospheric diffusion capacity,
and the neutralization capability of atmospheric aerosol have strong regional variations in China (Li,
1998). Although the spatial pattern of PA and precipitation pH were different, both of them showed the
acid pollution had decreased in 2016 even compared with 2005 and 2010.
**3.5 Uncertainties**
Since the ABL $SO_2$ and $NO_2$ could indicate the variations of $SO_4^{2-}$ and $NO_3^-$ in precipitation, the
uncertainty induced by OMI $NO_2$ columns should be considered in this study. The uncertainty in
satellite-based vertical columns is dominated by air mass factors, which have been discussed in detail



in a number of previous studies (Boersma et al., 2004;Nowlan et al., 2014;Zyrichidou et al., 2013).
Especially in the western China where the values of $SO_2$ and $NO_2$ were relative low, errors of the ABL
$SO_2$ and $NO_2$ columns might be high.
This study only considered the potential acidity induced by the $SO_4^{2-}$ and $NO_3^-$ the organic acid was not
involved. Although the contribution of organic acids to precipitation pH was minor, it could not be
neglected, particularly in forest and suburban areas (Stavrakou et al., 2012;Willey et al., 2011). In these
areas, the contribution of organic acids on precipitation pH was much higher than in urban areas.
This study mainly discussed the change of acidic species in precipitations, but the neutralization should
be fully considered when the precipitation pH is studied. The implementation of particulate matter
reduction policy has not only resulted in the decreasing trend of the acid-related compound, but also the
decrease of alkaline species in precipitation (Zhao et al., 2009;Wang et al., 2012;Tang et al., 2010).
However, the neutralizing is not considered in this study.
**4. Conclusions**
The effect of national $NO_x$ and $SO_2$ policies on acid components in precipitation was assessed in China,
based on the OMI ABL $SO_2$ and NO2 columns. The OMI information on S and N obtained a reliable
indicator on the variations of $SO_4^{2-}$ / $NO_3^-$ in precipitation. The long-term trend of S / N in precipitation
significantly decreased from 2005 to 2016 in China, which meant the contribution of nitrates getting
higher on precipitation pH. This decline of S / N in precipitation was mainly due to the national $NO_x$
and $SO_2$ policies enacted at different times. Under such policies, $SO_2$ showed a decreasing trend while
$NO_2$ showed the increasing first then decreasing trend from 2005 to 2016. The ABL $SO_2$ and $NO_2$
columns had a good consistency with those of emissions, but they were also greatly influenced by the
precipitation amounts. Particularly for the year of 2011, $SO_2$ and $NO_2$ got their peaks, respectively,



mainly due to the joint contribution from the lowest precipitation amounts and high emissions in 2011.
The spatial distribution of S / N in precipitation showed considerable regional variations in China. The
low values were mainly located in East China, and the areas around Urumqi in Northwest China, which
is close to the spatial distribution of population densities. The highly developed traffics in the densely
populated areas, and the more widespread application of the FGD than SCR in power units might be
the main reasons for the low S / N in the mentioned regions.
The potential acidity of S and N in precipitation increased first from 2005 to 2011 and then decreased
from 2011 to 2016. The increased PA indicated that the benefits of $SO_2$ reductions during 2005-2010
might be offset by increased N emissions obtained by the previous studies, while the decreased PA
from 2011 to 2016 indicated that the national $NO_x$ and $SO_2$ policies issued in 12th Five Plan are in
effect.
**Author contribution**
Xiuying Zhang conceived and designed the methodology and wrote the paper with Dongmei Chen. Lei
Liu, Limin Zhao, and Wuting Zhang help to process the data sets.
**Acknowledgements**
This study is supported by the National Natural Science Foundation of China (No. 41471343), and the
Fundamental Research Funds for the Central Universities (0904-14380013). We acknowledge the free
use of tropospheric $NO_2$ column and ABL $SO_2$ column from the OMI sensor from www.temis.nl. We
also thank Louisa Emmons from National Center for Atmospheric Research (NCAR) for providing the
MOZART output data.
**Competing interests**
The authors declare that they have no conflict of interest.





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






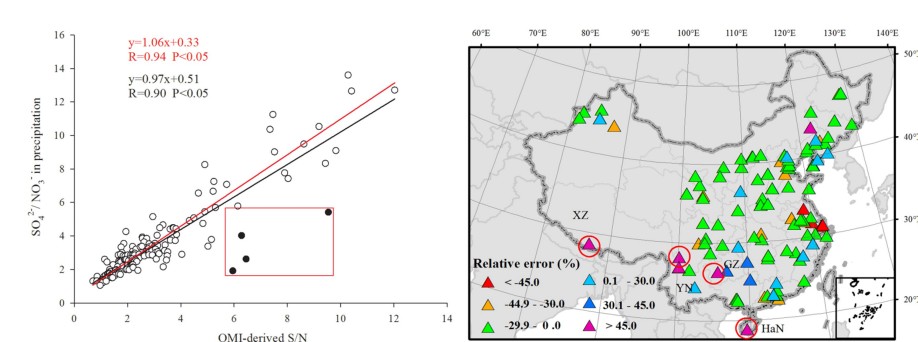

**Fig. 1.** (a) Scatter plots of the OMI-derived S / N and the collected $SO_4^{2-}$ / $NO_3^-$ in precipitation; (b) spatial distribution of the

578          relative errors between the OMI-derived S / N and the collected ratio of $SO_4^{2-}$ / $NO_3^-$ in precipitation




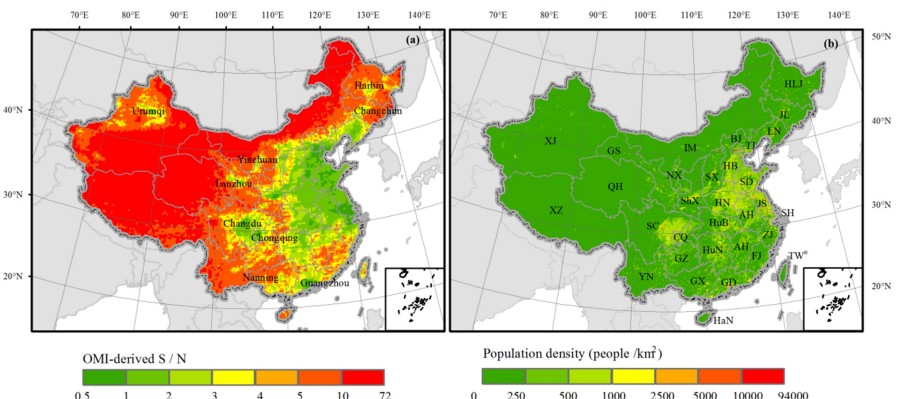

**Fig. 2.** Spatial distribution of (a) OMI-derived S / N in 2016 and (b) the population density in China. The successfully full

provincial names are Beijing (BJ), Tianjin (TJ), Hebei (HeB), Shandong (SD), Shanxi (SX), Henan (HeN), Shaanxi (SaX),

Liaoning (LN), Jilin (JL), Heilongjiang (HLJ), Neimenggu (NMG), Gansu (GS), Ningxia (NX), Xinjiang (XJ), Shanghai (SH),

Jiangsu (JS), Zhejiang (ZJ), Anhui (AH), Hubei (HuB), Hunan (HuN), Jiangxi (JX), Fujian (FJ), Guangdong (GD), Hainan

(HaN), Yunnan (YN), Guizhou (GZ), Chongqing (CQ), Sichuan (SC), Guangxi (GX), Xizang (XZ), Qinghai (QH), and Taiwan

(TW).





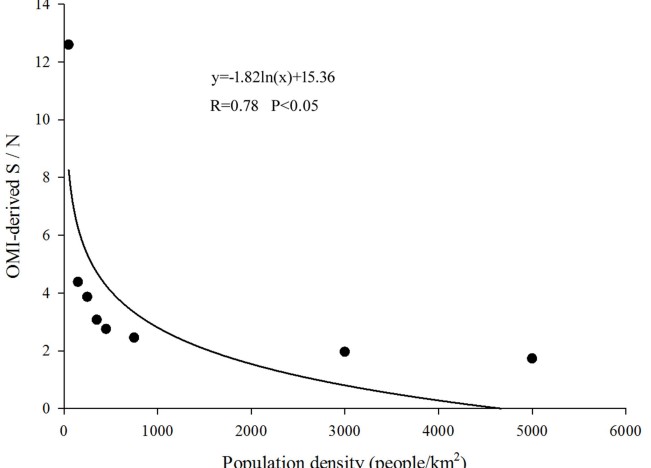

        **Fig. 3.** Relationship between Omi-derived S / N and population density





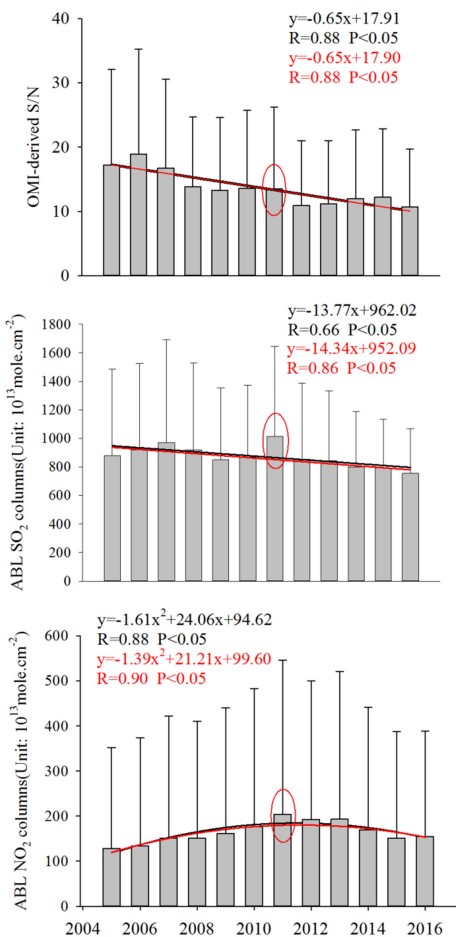


**Fig. 4.** Long-term trends of OMI-derived S / N, ABL SO$_2$ columns, ABL NO$_2$ columns from 2005 to 2016 in China



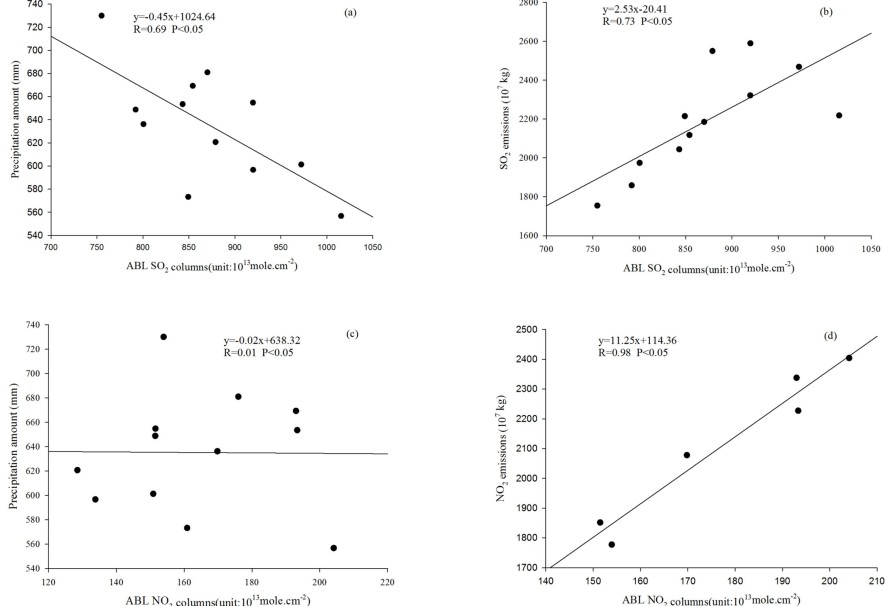


**Fig. 5.** Relationship between the (a) ABL $SO_2$ columns and precipitation amounts, (b) ABL $SO_2$ columns and $SO_2$ emissions, (c)

ABL $NO_2$ columns and precipitation amounts, (b) ABL $NO_2$ columns and NOx emissions






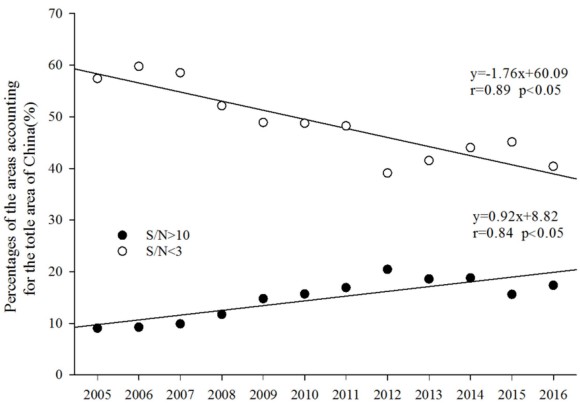


**Fig.6.** Percentages of the areas with mixed acidic precipitation types from 2005 to 2016





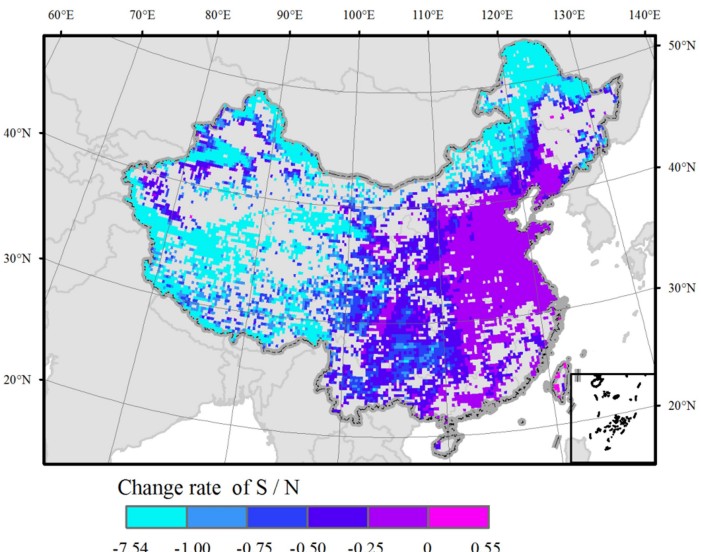


**Fig.7.** Linear trend per year for the ratio of OMI-derived S/ N in from 2005 to 2016 derived from OMI. For the light grey areas

600                                    no significant trend has been found in the time series.



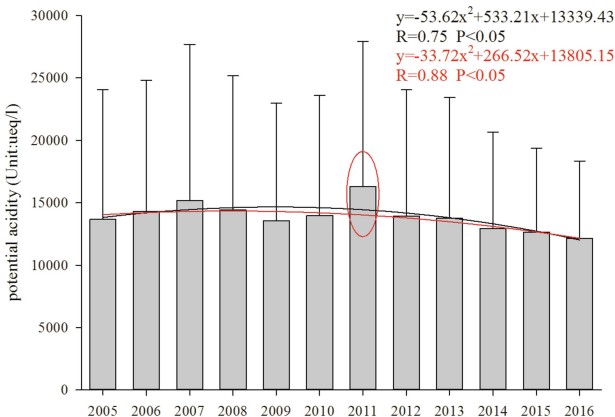

**Fig. 8.** Trends of potential acidity induced by $SO_4^{2-}$ and $NO_3^-$ from 2005 to 2016 in China





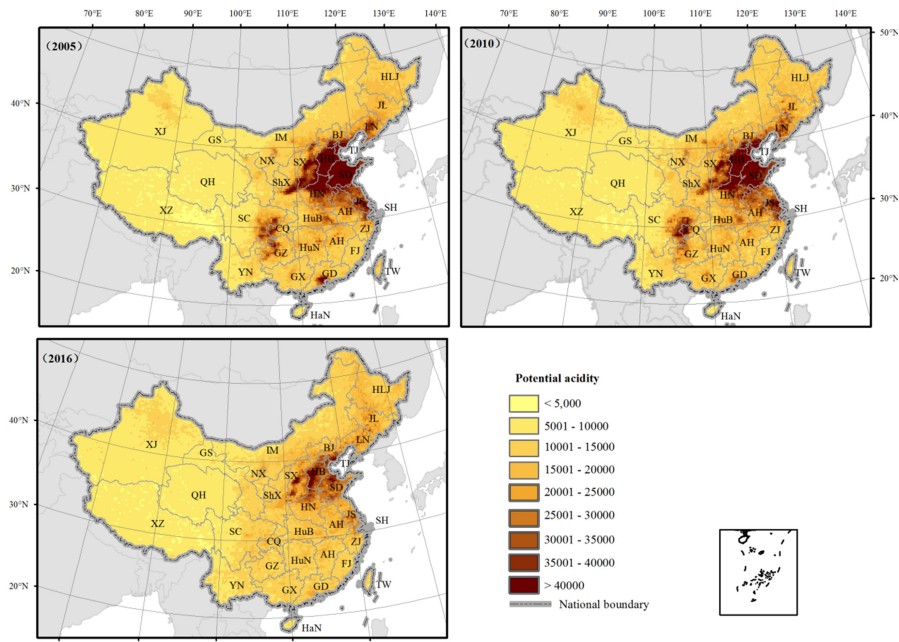


**Fig. 9**. Spatial distributions of potential acidity in 2005, 2010, and 2016 in China. The successfully full provincial names are the
same with **Fig. 2**.

