# Peer review of "Effect assessment of NOx and SO2 control policies on acid species in precipitation from 2005 to"

_Atmospheric Chemistry and Physics, 2017_

## Referee Comment (RC1) · Anonymous Referee #1 · 12 Oct 2017

Review of Zhang et al., "Effect assessment of NOx and SO2 control policies on acid species in precipitation from 2005 to 2016 in China based on satellite monitoring"

In this manuscript, Zhang et al., use OMI retrievals to assess the columnar S/N ratios over 12 years in China. They compare the ratios to a collection of observations of S/N ratios in precipitation.

The major strength of this manuscript is it reports on columnar S/N ratios and shows they are consistent with observational trends. That is also its weakness as it does not go sufficiently beyond that to warrant publication at this time in ACP. It would be better suited to a journal that is more policy focused as it does not provide significant

insights to atmospheric chemistry or physics processes. Further, it is, at present, too long given the limited message of the manuscript. (At times, it appears that steps are taken to lengthen the article needlessly.) Another major issue is that the grammar is also not up to ACP standards, and significant editing will be required. Finally, the title is a bit misleading. The major thrust of the article is to use OMI S and N columnar abundances to assess how the ratio of those columns change, and the relationship to precipitation is done via other observations. Not surprisingly, the trends in the columnar abundances have a similar trend to the observations.

Digging in to the details a bit, the precipitation chemistry data set needs to be better described and documented., particularly discussing the spatial and temporal observational limitations. From what I can tell, much of the evaluation is based on the 474 standard ground observations. This is comparison is probably one of the key take-homes from the article at present. It is not apparent how those 474 locations compare to the "168" records related to the cited Liu et al., publications. This was confusing.

They need to better define their terms and show what was done with respect to the evaluation. They state "Other parameters of relative error (RE) and absolute error (AE) are used to assess the accuracy of the estimated NO2 by the following function:" but no function(s) are given. They also never show the AE. Further, it is not really apparent what should be taken from their performance evaluation.

As noted, the grammar is rough. For example, each sentence in the first paragraph is incorrect grammatically. I am still not certain what is meant by the "successfully" in "successfully full provincial names" means.

The appropriate term is "evaluation", not "validation" (throughout). Further, this study does not "confirm", but "supports" other findings.

They use "might" and other similar terms excessively, probably avoiding saying something more definitive.

They state "Although the contribution of organic acids to precipitation pH was minor, it could not be neglected, particularly in forest and suburban areas (Stavrakou et al., 2012;Willey et al., 2011)." First, they don't show it is minor. Next, what is meant by "could not".

The arbitrary classification of "sulfuric" and "mixed" acid precipitation does not add to the article, and related discussion can be dropped throughout. First, the demarcation is arbitrary. Second, the trend is the split does not add to our understanding of the chemistry or physics. It adds text.

In terms of other places where excessive text appears to be added, there are often long strings of province names. This hurts readability and adds little to the knowledge that is to be transferred. If there are specific characteristics, those should be discussed (look at the paragraph beginning at line 233.)

This article would be more fitting as a note, again more appropriately in a journal more aligned with air quality policy, greatly shortened and retitled (e.g., Trends in OMI S and N columns and comparison to trends in precipitation composition"). The analysis would show the S and N and S/N trends, and the ratio (The current Fig. 4). In addition to being tightened and more focused, the grammar also needs to be addressed before further consideration by this, or other, journals.

---

## Referee Comment (RC2) · Anonymous Referee #2 · 2 Dec 2017

**Review of Zhang et al. for Atmospheric Chemistry and Physics**

*General Comments*

In "Effect assessment of $NO_x$ and $SO_2$ control policies on acid species in precipitation from 2005 to 2016 in China based on satellite monitoring", Zhang et al. describe a metric based on a correlation of OMI observations of $NO_2$ and $SO_2$ with wet deposited nitrate and sulfate. The statistical coefficient from each correlation is multiplied by an OMI-derived "atmospheric boundary layer" column concentration, and the ratio of these values is said to represent the ratio of sulfate to nitrate ions in precipitation. They present correlations of this "OMI-derived S/N" metric with sulfate to nitrate ratios measured in precipitation, time (2005-2016), and population density (2010). The authors develop a "potential acidity" metric from the same statistical coefficient that was derived from the observations. The aims of the paper seem to be to define a satellite-based metric to identify the acidic constituents of precipitation and to use it to affirm that emissions control policies have reduced the amount of sulfate in precipitation more than nitrate.

The authors begin with a thorough description of emissions policies in China over the last two decades. The methods rely heavily on assertions from linear correlations. The data the authors analyze is not well documented. Some of the data, including the measured sulfate and nitrate (Section 2.1.3) are referenced to other published papers but without a brief characterization of the nature of the data collection or extent spatially or temporally of the dataset. Other key parameters in the statistical models are not described, such as the atmospheric boundary layer $SO_2$ term. Evidence of the quality of the models (eq. 5) essential to calculating the OMI-derived S/N metric is not presented. Although the model for nitrogen deposition was recently published in another journal, the model for sulfate deposition, from which the important correlation coefficient arises for the S/N metric, has not yet been peer reviewed. The explanations of data included or excluded in figures (e.g., very small number of observations in Fig 5c) is neglected as are the sources of the emissions. Throughout the manuscript, the quality of the statistical analysis, representations of error, and propensity to neglect outliers without physical reason calls into question the quality of these underlying statistical models, which are not evaluated in this manuscript.

Furthermore, the authors seek to explain an outlier in the $SO_2$ downward trend by acknowledging that 2011 had the least precipitation on record for the last 50 years. The authors do not take this opportunity to acknowledge the inherent difficulty in using a gas phase satellite-based observation of precipitable species to explain wet deposition nor do they explain how the linear models that are the basis for the S/N metric account for the way that rain depletes the nitrate and sulfate concentrations in the atmosphere. Finally, the authors do not make a case for using a satellite-based metric for estimating the ratio of wet-deposited sulfate to nitrate when measurements of the ions in rain water are already being conducted across China. Because of the poor evaluation of the statistical models underlying the conclusions, the absence of documentation of datasets, the inherent difficulty in the approach attempted, and the lack of purpose for the results, I cannot recommend this manuscript for publication in Atmospheric Chemistry and Physics.

---

## Author Comment (AC1) · 10 Feb 2018

1. The major strength of this manuscript is it reports on columnar S/N ratios and shows they are consistent with observational trends. That is also its weakness as it does not go sufficiently beyond that to warrant publication at this time in ACP. It would be better suited to a journal that is more policy focused as it does not provide significant insights to atmospheric chemistry or physics processes. Further, it is, at present, too long given the limited message of the manuscript. (At times, it appears that steps are taken to lengthen the article needlessly.) Another major issue is that the grammar is also not up to ACP standards, and significant editing will be required. Finally, the title

is a bit misleading. The major thrust of the article is to use OMI S and N columnar abundances to assess how the ratio of those columns change, and the relationship to precipitation is done via other observations. Not surprisingly, the trends in the columnar abundances have a similar trend to the observations.

Response: We highly appreciate the reviewer's comments. In order to make our manuscript better fit the scope of ACP, we have made significant changes on data and analysis of our study. In this round of revision, we have collected the ground measurements on $SO_4^{2-}$ and $NO_3^-$ concentrations in precipitations from 2005 to 2016 at 60 sites across China. We used the ground measurements to construct the model to estimate wet $SO_4^{2-}$ -S and $NO_3^-$ -N depositions, based on the SO2 and NO2 columns in atmospheric boundary layer (ABL). The rationale behind this is that NO2 reacts with O3 to form $NO_3-$ and then highly soluble N2O5 ($NO_2 + NO_3 \rightarrow N_2O_5$, $N_2O_5 + H_2O \rightarrow 2\ HNO_3$), thus most of the bulk $NO_3^-$-N in precipitation originates from HNO3 and aerosol nitrate ($NO_3-$) (Barrie, 1985;Liu et al., 2017). Similarly, the rationale of using SO2 columns to estimate wet S deposition is based on the relationship between SO2 and $SO_4^{2-}$. At the gas phase, SO2 is oxidized by reaction with the hydroxyl radical via an intermolecular reaction ($SO_2 + OH\hat{a}\check{A}\acute{c} \rightarrow HOSO_2\hat{a}\check{A}\acute{c}$), which is followed by ($HOSO_2\hat{a}\check{A}\acute{c} + SO_2 \rightarrow HO_2\hat{a}\check{A}\acute{c} + SO_3$); in the presence of water, sulfur trioxide (SO3) is converted rapidly to sulfuric acid ($SO_3\ (g) + H_2O\ (l) \rightarrow H_2SO_4$). Furthermore, the wet deposition flux (F) could be estimated by $F = W \times P \times C$, where W is scavenging ratios, P is precipitation amount, and C indicates SO2 or NO2 concentrations in atmosphere (Barrie, 1985;Sakata et al., 2006). We used the ground on $SO_4^{2-}$ or $NO_3^-$ concentrations and the ABL SO2 or NO2 to estimate the scavenging ratios.

The results showed that SO2 and NO2 columns in ABL have potential to estimate wet $SO_4^{2-}$ -S and $NO_3^-$ -N depositions (R = 0.883, intercept = 0.903, P < 0.05 for $SO_4^{2-}$–S estimations; and R = 0.893, intercept = 0.755, P < 0.05 for $NO_3^-$–N estimations). In this version, we have shifted the focus on discussing the trends of wet $SO_4^{2-}$ -S and $NO_3^-$ -N depositions across China rather than the effect of the air quality polices.

We have tried to polish the language and improve the writing quality of the manuscript. However, if it is still not up to ACP standards, we will send it to professional language editing.

In this version, we have used the wet SO42- -S and NO3- -N depositions to compare the contributions of SO42- and NO3- on precipitation acidity, and further to detect the potential acidity (PA) induced by H2SO4 and HNO3. We have developed the models of linking the trends in S/N and PA with the precipitation amount and ABL SO2 and NO2 columns. This study has generated much fine spatial maps of S/N and PA distribution which can not be achieved by only ground observations.

In this version, we have changed the title "Effect assessment of NOx and SO2 control policies on acid species in precipitation from 2005 to 2016 in China based on satellite monitoring" to "Contribution ratio and trends of SO42- to NO3- to precipitation acidity from 2006 to 2016 in China based on OMI observations", which more focuses on the content of the manuscript.

2. Digging in to the details a bit, the precipitation chemistry data set needs to be better described and documented. Particularly discussing the spatial and temporal observational limitations. From what I can tell, much of the evaluation is based on the 474 standard ground observations. This is comparison is probably one of the key take homes from the article at present. It is not apparent how those 474 locations compare to the "168" records related to the cited Liu et al., publications. This was confusing.

Response: In this revision, we have used the ground measurements on SO42- and NO3- concentrations in precipitations to replace the data collected from published papers. The precipitation samples at 60 sites are collected based on the routine procedure on acid precipitation monitoring technology (HJ/T165ïij■2004). The quality of monitoring data is evaluated and supervised by China National Accreditation Board for Laboratories according to international requirements. The detailed information on the locations of sites, number of the collected precipitation events and spanned time is

listed in Table S1.

In the previous version, we estimated the S/N across China through the constructed model based on ABL SO2, ABL NO2, and SO42- / NO3- in the 168 data records from the published papers. We extracted the S/N values from the estimated S/N map, and got the estimated average S/N at 474 sites. In this version, this accuracy assessment has been replaced by that of the outputs from the constructed models with the ground measurements.

3. They need to better define their terms and show what was done with respect to the evaluation. They state "Other parameters of relative error (RE) and absolute error (AE) are used to assess the accuracy of the estimated NO2 by the following function:" but no function(s) are given. They also never show the AE. Further, it is not really apparent what should be taken from their performance evaluation.

Response: In this version the function of RE is listed as the Eq. (2) and AE is not used. We used the average of RE to evaluate the accuracy of the estimated wet SO42- -S and NO3- -N depositions.

4. As noted, the grammar is rough. For example, each sentence in the first paragraph is incorrect grammatically. I am still not certain what is meant by the "successfully" in "successfully full provincial names" means. The appropriate term is "evaluation", not "validation" (throughout). Further, this study does not "confirm", but "supports" other findings. They use "might" and other similar terms excessively, probably avoiding saying something more definitive.

Response: Thank you for helping us improve the writing quality of our paper. In this version, we have tried to improve the grammar in this revision.

5. They state "Although the contribution of organic acids to precipitation pH was minor, it could not be neglected, particularly in forest and suburban areas (Stavrakou et al., 2012; Willey et al., 2011)." First, they don't show it is minor. Next, what is meant by

"could not".

Response: In fact, we just want to discuss the potential uncertainty caused by not considering the contribution of organic acids in this study. In this version, we have changed this sentence to "This study only considered the potential acidity induced by the H2SO4 and HNO3, the organic acid was not considered due to its minor contribution on precipitation acidity (Stavrakou et al., 2012;Willey et al., 2011)."

6. The arbitrary classification of "sulfuric" and "mixed" acid precipitation does not add to the article, and related discussion can be dropped throughout. First, the demarcation is arbitrary. Second, the trend is the split does not add to our understanding of the chemistry or physics. It adds text.

Response: In this version the classification and the related discussions of "sulfuric" and "mixed" acid precipitation have been removed.

7. In terms of other places where excessive text appears to be added, there are often long strings of province names. This hurts readability and adds little to the knowledge that is to be transferred. If there are specific characteristics, those should be discussed (look at the paragraph beginning at line 233.)

Response: We have changed the full name of the provinces to their logograms. We have also cut some excessive texts to make the discussion more concise and focused.

8. This article would be more fitting as a note, again more appropriately in a journal more aligned with air quality policy, greatly shortened and retitled (e.g., Trends in OMI S and N columns and comparison to trends in precipitation composition"). The analysis would show the S and N and S/N trends, and the ratio (The current Fig. 4). In addition to being tightened and more focused, the grammar also needs to be addressed before further consideration by this, or other, journals.

Response: Please refer to our detailed response for #1. In this version, we have shifted the focus from discussing the effect of the air quality polices to the trends of wet SO42-

-S and NO3- -N depositions across China as the reviewer suggested. We have used the wet SO42- -S and NO3- -N depositions to compare the contributions of SO42- and NO3- on precipitation acidity, and further to detect the potential acidity (PA) induced by H2SO4 and HNO3.

The length of the manuscript has been shortened, from the word number of "8306" in the previous version to "6618" in this version.

The core model to estimate the wet SO42- -S and NO3- -N depositions in this study is a statistical model combining ground measurements and satellite observations. This estimation had the advantages of high spatial resolution. We resolved the problem of wet deposition from another view point apart from the atmospheric chemical transport (ACT) theory, and the key point is that we gained a reliable result. This is our contribution.

Reference:

Barrie, L. A.: Scavenging ratios, wet deposition, and in-cloud oxidation-an application to the oxides of sulfur and nitrogen, Journal of Geophysical Research-Atmospheres, 90, 5789-5799, 10.1029/JD090iD03p05789, 1985.

Liu, L., Zhang, X., Xu, W., Liu, X., Lu, X., Chen, D., Zhang, X., Wang, S., and Zhang, W.: Estimation of monthly bulk nitrate deposition in China based on satellite NO2 measurement by the Ozone Monitoring Instrument, Remote Sensing of Environment, 199, 14, 2017.

Sakata, M., Marumoto, K., Narukawa, M., and Asakura, K.: Regional variations in wet and dry deposition fluxes of trace elements in Japan, Atmospheric Environment, 40, 521-531, 10.1016/j.atmosenv.2005.09.066, 2006.

Stavrakou, T., Mueller, J. F., Peeters, J., Razavi, A., Clarisse, L., Clerbaux, C., Coheur, P. F., Hurtmans, D., De Maziere, M., Vigouroux, C., Deutscher, N. M., Griffith, D. W. T., Jones, N., and Paton-Walsh, C.: Satellite evidence for a large source of formic

acid from boreal and tropical forests, Nature Geoscience, 5, 26-30, 10.1038/ngeo1354, 2012.

Willey, J. D., Glinski, D. A., Southwell, M., Long, M. S., Avery, G. B., Jr., and Kieber, R. J.: Decadal variations of rainwater formic and acetic acid concentrations in Wilmington, NC, USA, Atmospheric Environment, 45, 1010-1014, 10.1016/j.atmosenv.2010.10.047, 2011.
* * *

---

## Author Comment (AC2) · 10 Feb 2018

1. The authors begin with a thorough description of emissions policies in China over the last two decades.

Response: We highly appreciate the reviewer's comments. This study aims to detect the trend of the contribution of $SO_4^{2-}$ and $NO_3^-$ on precipitation acidity, which is greatly influenced by the policies enacted in China to improve air quality. Therefore, we gave the background in the Introduction. In this version, we have shifted the focus from discussing the effect of the air quality polices to the trends of wet $SO_4^{2-}$ -S and $NO_3^-$ -N depositions across China as the reviewer suggested.

[Figure]

2. The methods rely heavily on assertions from linear correlations.

Response: In order to make our manuscript better fit the scope of ACP, we have made significant changes on data and analysis of our study. In this round of revision, we have collected the ground measurements on SO42- and NO3- concentrations in precipitations from 2005 to 2016 at 60 sites across China. We used the ground measurements to construct the model to estimate wet SO42- -S and NO3- -N depositions, based on the SO2 and NO2 columns in atmospheric boundary layer (ABL). The rationale behind this is that NO2 reacts with O3 to form NO3− and then highly soluble N2O5 (NO2 + NO3 → N2O5, N2O5+H2O→2 HNO3), thus most of the bulk NO3−-N in precipitation originates from HNO3 and aerosol nitrate (NO3−) (Barrie, 1985;Liu et al., 2017). Similarly, the rationale of using SO2 columns to estimate wet S deposition is based on the relationship between SO2 and SO42-. At the gas phase, SO2 is oxidized by reaction with the hydroxyl radical via an intermolecular reaction (SO2 + OH• → HOSO2•), which is followed by (HOSO2• + SO2 → HO2• + SO3); in the presence of water, sulfur trioxide (SO3) is converted rapidly to sulfuric acid (SO3 (g) + H2O (l) → H2SO4). Furthermore, the wet deposition flux (F) could be estimated by F=W×P×C, where W is scavenging ratios, P is precipitation amount, and C indicates SO2 or NO2 concentrations in atmosphere (Barrie, 1985;Sakata et al., 2006). We used the ground on SO42- or NO3- concentrations and the ABL SO2 or NO2 to estimate the scavenging ratios.

The results showed that SO2 and NO2 columns in ABL have potential to estimate wet SO42- -S and NO3- -N depositions (R = 0.883, intercept = 0.903, P < 0.05 for SO42–S estimations; and R = 0.893, intercept = 0.755, P < 0.05 for NO3–N estimations).

Therefore, although the core model to estimate the wet SO42- -S and NO3- -N depositions in this study is a statistical model, it has rationale behind. We resolved the problem of wet deposition from another view point apart from the atmospheric chemical transport (ACT) theory, and the key point is that we gained a reliable result.

3. The data the authors analyze is not well documented. Some of the data, including

the measured sulfate and nitrate (Section 2.1.3) are referenced to other published papers but without a brief characterization of the nature of the data collection or extent spatially or temporally of the dataset.

Response: In this revision, we have used the ground measurements on $SO_4^{2-}$ and $NO_3^-$ concentrations in precipitations to replace the data collected from published papers. The precipitation samples at 60 sites are collected based on the routine procedure on acid precipitation monitoring technology (HJ/T165ïij■2004). Precipitation amount is measured immediately after every precipitation event finished, and the concentrations of $SO_4^{2-}$ and $NO_3^-$ are measured using ion chromatography. The quality of monitoring data is evaluated and supervised by China National Accreditation Board for Laboratories according to international requirements. The detailed information on the locations of sites, number of the collected precipitation events and spanned time is listed in Table S1.

4. Other key parameters in the statistical models are not described, such as the atmospheric boundary layer $SO_2$ term. Evidence of the quality of the models (eq. 5) essential to calculating the OMI-derived S/N metric is not presented. Although the model for nitrogen deposition was recently published in another journal, the model for sulfate deposition, from which the important correlation coefficient arises for the S/N metric, has not yet been peer reviewed.

Response: Please refer to our detailed response for #2.

5. The explanations of data included or excluded in figures (e.g., very small number of observations in Fig 5c) is neglected as are the sources of the emissions. Throughout the manuscript, the quality of the statistical analysis, representations of error, and propensity to neglect outliers without physical reason calls into question the quality of these underlying statistical models, which are not evaluated in this manuscript.

Response: In this version, the analysis on Fig. 5 has been removed. And we have tried to correct the questions mentioned above.

6. Furthermore, the authors seek to explain an outlier in the SO2 downward trend by acknowledging that 2011 had the least precipitation on record for the last 50 years.

Response: In fact, SO2 columns showed a downward from 2005 to 2016, except for the year of 2011. Since SO2 emissions did not high in this year, the low removal by dry and wet depositions and the long-range transport might be the reasons. Therefore, our analysis in the previous version does not fully consider all of the factors. In this version, the analysis on the peak of SO2 in 2011 has been removed, since the related contents have been removed.

7. The authors do not take this opportunity to acknowledge the inherent difficulty in using a gas phase satellite based observation of precipitable species to explain wet deposition nor do they explain how the linear models that are the basis for the S/N metric account for the way that rain depletes the nitrate and sulfate concentrations in the atmosphere.

Response: Please refer to our detailed response for #2.

8. Finally, the authors do not make a case for using a satellite-based metric for estimating the ratio of wet-deposited sulfate to nitrate when measurements of the ions in rain water are already being conducted across China. Because of the poor evaluation of the statistical models underlying the conclusions, the absence of documentation of datasets, the inherent difficulty in the approach attempted, and the lack of purpose for the results, I cannot recommend this manuscript for publication in Atmospheric Chemistry and Physics.

Response: Although the acid rain pollution across China has been monitored, not all of the chemical compositions in precipitations have been measured. Furthermore, the report on the spatial distribution of the wet $SO_4^{2-}$ -S and $NO_3^-$ -N depositions based on ground measurements has not been reported yet. The key contributions of this study are that we estimate the long-term trend of wet $SO_4^{2-}$ -S and $NO_3^-$ -N depositions with high spatial resolutions across China, compare the contributions of H2SO4 and

[Figure]

HNO3 on precipitation acidity, and detect the trend in their combining acidity. The trend of acid species in precipitations would help understand the acid pollution in China and thus make efficient way to control acid pollutions. We believe the results in this study will provide the scientific information to the experts in global change studies and the policy makers.

Reference:

Barrie, L. A.: Scavenging ratios, wet deposition, and in-cloud oxidation-an application to the oxides of sulfur and nitrogen, Journal of Geophysical Research-Atmospheres, 90, 5789-5799, 10.1029/JD090iD03p05789, 1985.

Liu, L., Zhang, X., Xu, W., Liu, X., Lu, X., Chen, D., Zhang, X., Wang, S., and Zhang, W.: Estimation of monthly bulk nitrate deposition in China based on satellite NO2 measurement by the Ozone Monitoring Instrument, Remote Sensing of Environment, 199, 14, 2017.

Sakata, M., Marumoto, K., Narukawa, M., and Asakura, K.: Regional variations in wet and dry deposition fluxes of trace elements in Japan, Atmospheric Environment, 40, 521-531, 10.1016/j.atmosenv.2005.09.066, 2006.